# The UKSCAPE-G2G river flow and soil moisture datasets: Grid-to-Grid model estimates for the UK for historical and potential future climates

Alison L Kay, Victoria A Bell, Helen N Davies, Rosanna A Lane, Alison C Rudd

UK Centre for Ecology & Hydrology, Wallingford, UK, OX10 8BB

Correspondence to: A.L. Kay (alkay@ceh.ac.uk)

## Abstract

Appropriate adaptation planning is contingent upon information about the potential future impacts of climate change, and hydrological impact assessments are of particular importance. The UKSCAPE-G2G datasets were produced, as part of the NERC UK-SCAPE programme, to contribute to this information requirement. They use the Grid-to-Grid (G2G) national-scale hydrological model configured for both Great Britain and Northern Ireland (and the parts of the Republic of Ireland that drain to rivers in NI). Six separate datasets are provided, for two sets of driving data — one observation-based (1980–2011) and one climate projection-based (1980–2080) — for both river flows and soil moisture on 1 km x 1 km grids across GB and NI. The river flow datasets include grids of monthly mean flow, annual maxima of daily mean flow, and annual minima of 7-day mean flow ($m^3s^{-1}$). The soil moisture datasets are grids of monthly mean soil moisture content (m water / m soil), which should be interpreted as depth-integrated values for the whole soil column. The climate projection-based datasets are produced using data from the 12-member 12km regional climate model ensemble of the latest UK climate projections (UKCP18), which uses RCP8.5 emissions. The production of the datasets is described, along with details of the file format, and how the data should be used. Example maps are provided, as well as simple UK-wide analyses of the various outputs. These suggest potential future decreases in summer flows, annual minimum 7-day flows, and summer/autumn soil moisture, along with possible future increases in winter flows and annual maximum flows. References are given for published papers providing more detailed spatial analyses, and some further potential uses of the data are suggested.

## Keywords

Climate change; hydrological impacts; rainfall-runoff; UK Climate Projections 2018; UKCP18

## 1  Introduction

Information on the potential future impacts of climate change is crucial to enable appropriate adaptation planning, and impacts on the hydrological cycle and river flows are one of the main ways by which climate change will affect both society and the natural environment. UK-SCAPE (UK Status, Change And Projections of the Environment; ukscape.ceh.ac.uk) is a five-year programme funded by the Natural Environment Research Council (NERC) as part of a National Capability Science Single Centre award, and the main aim of Work Package 2.2 of UK-SCAPE is to

deliver data and analyses showing how future climate change could influence water
quantity. The hydrological datasets presented here were produced as part of UK-
SCAPE WP2.2.
The datasets consist of 1 km x 1 km gridded outputs from a national-scale
hydrological model (Grid-to-Grid), and include both river flows and soil moisture, for
Great Britain (GB) and Northern Ireland (NI). The model has been driven with
observation-based data, and with an ensemble of Regional Climate Model (RCM)
data from the latest climate projections for the UK (UKCP18; Lowe et al. 2018). A
summary of the six available datasets, including references, is provided in Table 1.
The datasets have been used within UK-SCAPE WP2.2 to support analyses of
potential future changes in river flows and soil moisture (Kay 2021, Lane & Kay
2021, Kay et al. 2021a, Kay et al. 2022a), but could also be used to support other
hydrological research and wider studies such as ecological and agricultural
modelling.
Section 2 describes how the datasets were produced, including the hydrological
model and the driving data applied. Section 3 presents some high-level analyses of
the datasets, and describes the results of more detailed analyses presented in other
published papers. Section 4 discusses potential uses and caveats, with conclusions
in Section 5.

**Table 1 Summary of the six UKSCAPE-G2G datasets.**

| **Observation-driven** | | |
|---|---|---|
| River flow | GB: | https://doi.org/10.5285/2f835517-253e-4697-b774-ab6ff2c0d3da (Kay et al. 2021c) |
| | NI: | https://doi.org/10.5285/f5fc1041-e284-4763-b8b7-8643c319b2d0 (Kay et al. 2021d) |
| Soil moisture | GB and NI: | https://doi.org/10.5285/c9a85f7c-45e2-4201-af82-4c833b3f2c5f (Kay et al. 2021e) |
| **Climate projection-driven** | | |
| River flow | GB: | https://doi.org/10.5285/18be3704-0a6d-4917-aa2e-bf38927321c5 (Kay et al. 2022b) |
| | NI: | https://doi.org/10.5285/76057b0a-b18f-496f-891c-d5b22bd0b291 (Kay et al. 2022c) |
| Soil moisture | GB and NI: | https://doi.org/10.5285/f7142ced-f6ff-486b-af33-44fb8f763cde (Kay et al. 2022d) |


# 2 Data production methods

## 2.1 The hydrological model

The Grid-to-Grid (G2G) is a national-scale grid-based hydrological model which
typically operates on a 1km x 1km grid at a 15-minute time-step (Bell et al. 2009),
with an optional snow module (Bell et al. 2016). It was originally configured to cover
Great Britain (GB), on a spatial domain aligned with the GB national grid, but more
recently a version was configured to cover Northern Ireland (NI) and areas in the
Republic of Ireland (RoI) that drain into NI, also on a domain aligned with the GB
national grid (Kay et al. 2021a). The G2G is configured using spatial datasets (e.g.
soil types, land-cover, flow directions), in preference to parameter identification via
calibration; where model parameters are required (such as the wave speeds used in
lateral routing) nationally-applicable values are applied (Bell et al. 2009).
G2G has been shown to perform well for a wide range of catchments across GB and
NI, including for the modelling of high flows / floods and low flows / droughts (Bell et
al. 2009, 2016; Rudd et al. 2017; Formetta et al. 2018; Kay et al. 2021a,b). This is
particularly the case for catchments with more natural flow regimes, as the model
does not routinely account for artificial influences like abstractions/discharges
(Rameshwaran et al. 2022). While the effect of urban/suburban land-cover on runoff
is accounted for, the effect of lake/reservoir storage and regulation is generally
neglected at the national scale; lake grid-cells are treated as though they were rivers.
This has a minimal effect across most of GB; the largest lake in Scotland, Loch
Lomond, has an area of ~71km$^2$, and the largest lake in England, Windermere, has
an area of ~15km$^2$. But in NI the dominant presence of Lough Neagh (~390km$^2$)
limits model performance downstream (the Lower Bann river; Kay et al. 2021a), and
Lough Erne in the south-west of NI is also relatively large (Upper and Lower Lough
Erne have a combined area of ~144km$^2$).
**2.2   Observation-based driving data**
Gridded time-series of precipitation and potential evaporation (PE) are required to
drive the G2G, plus temperature for the snow module. The observation-based driving
data are applied as follows:
• Daily 1km grids of precipitation (CEH-GEAR; Tanguy et al. 2016) are divided
equally over each model time-step within a day. For use in NI they are first re-
projected from the Irish national grid to the GB national grid.
• Monthly 40km grids of PE for short grass (MORECS; Hough and Jones 1997) are
copied down to the 1km grid, then divided equally over each model time-step
within a month. For use in NI they are first re-projected from the Irish national grid
to the GB national grid. The data do not cover all the required parts of the UK, so
have been extended where necessary (i.e. some coastal areas and some parts of
the RoI that drain into NI) by copying from the nearest cell with data.
• Daily 1km grids of min and max temperature (HadUK-Grid; Met Office 2019) are
interpolated through the day using a sine curve (Kay and Crooks 2014). The data
do not cover the required parts of the RoI, so have been infilled from the nearest
cell with data, using a lapse rate with elevation data (Morris and Flavin 1990).
**2.3   Climate projection-based driving data**
The climate change simulations use data from the UKCP18 Regional projections
(Met Office Hadley Centre 2018). These comprise a 12-member perturbed
parameter ensemble (PPE) of the Hadley Centre ~12km Regional Climate Model
(RCM) nested in an equivalent PPE of their ~60km Global Climate model (GCM)
(Murphy et al. 2018). Ensemble member 01 represents the standard
parameterisation, with members 02-15 representing a range of credible variations in
parameters (note that there are no RCM equivalents for GCM PPE members 02, 03
and 14). The data cover Dec 1980–Nov 2080 under just one emissions scenario,
RCP8.5 (Riahi et al. 2011), and have a 360-day year (twelve 30-day months). The
data are available re-projected from the native climate model grid onto a 12km grid
aligned with the GB national grid – the latter are used here.
The climate projection data are applied as follows:
• Daily 12km grids of precipitation are directly available from the UKCP18 Regional
projections. These are first adjusted for bias using 12km grids of monthly
correction factors derived by comparing baseline values against CEH-GEAR data
averaged up to the 12km resolution (Kay 2021, Kay et al. 2021a). They are then
spatially downscaled to 1km using patterns of average annual rainfall (1961–
1990; Bell et al. 2007), and divided equally over each model time-step within a
day (as for observed data).
• Daily 12km grids of PE are not directly available from the UKCP18 Regional
projections. Instead, they are calculated from other meteorological variables in a
way which closely replicates MORECS (as in Robinson et al. (2021, 2022), but
using the bias-adjusted precipitation in the interception component). PE is only
estimated for 12km 'land' RCM boxes; where PE is required for boxes classed as
'sea' in the RCM, it is copied from the nearest 12km 'land' box. The method also
includes increased stomatal resistance under future higher atmospheric $CO_2$
concentrations (Rudd and Kay 2016, Guillod et al. 2018). The 12km PE are
copied down to the 1km grid, then divided equally over each model time-step
within a month (as for observed data).
• Daily 12km grids of min and max temperature are directly available from the
UKCP18 Regional projections. These are downscaled to 1km using a lapse rate
with elevation data, and interpolated through the day using a sine curve (as for
observed data).
**2.4 Hydrological model runs and outputs**
The observation-based simulation (hereafter 'SIMOBS') is initialised using a states
file saved at the end of a prior observation-based run (Jan 1970–Nov 1980). The
same state initialisation file is used for each RCM-based simulation (hereafter
'SIMRCM').
Model outputs consist of 1km x 1km gridded time-series of
• monthly mean river flow ($m^3s^{-1}$);
• annual maxima (AMAX) of daily mean river flow ($m^3s^{-1}$), for water years
(October–September);
• annual minima (AMIN) of 7-day mean river flow ($m^3s^{-1}$), for years spanning
December–November; and
• monthly mean soil moisture content (m water / m soil).
While it is possible to output 1km x 1km gridded daily time-series from G2G, these
are not typically produced as they are very large files (especially if long time periods
are covered, as is the case for the SIMRCM runs). Instead, the AMAX and AMIN
flows are calculated and saved during the model run, to enable analyses of high and
low flows without saving daily gridded flows. The AMAX of daily mean flows are
extracted for water years (Oct–Sep), to try to avoid extraction of the same high flow
event from consecutive years. AMIN extraction would usually use calendar years,
but Dec-Nov is used here to match the climate model data running from December
1980 to November 2080, whilst still trying to avoid extraction of the same low flow
event from consecutive years.
The flow variables are provided for all non-sea and non-tidal 1km cells with a
catchment drainage area of at least $50km^2$, while the soil moisture is provided for all
non-sea 1km cells. G2G soil moisture estimates are provided as monthly averages
of daily mean soil moisture in the unsaturated zone, which can be interpreted as
volumetric soil moisture content, θ, where $0 \leq θ \leq 1$. In G2G soil depth can vary from
a few centimetres to several metres, and soil moisture estimates should be
interpreted as depth-integrated values for the whole soil column.

## 2.5   Format of the gridded datasets

The 1km x 1km gridded data are provided as a NetCDF4 file for each variable,
following UKCEH gridded dataset conventions. The file naming convention is
described in Table 2 for the observation-based datasets and Table 3 for the climate
projection-based datasets.

**Table 2 The file naming convention for the observation-based datasets.**

| Data | Names of NetCDF files | Years available |
|------|----------------------|-----------------|
| monthly mean river flow | G2G_GB_mmflow_obs_1980_2011.nc<br>G2G_NI_mmflow_obs_1980_2011.nc | Dec 1980–Nov 2011 |
| annual maxima of daily mean river flow | G2G_GB_amaxflow_obs_1980_2011.nc<br>G2G_NI_amaxflow_obs_1980_2011.nc | Oct 1981–Sep 2011 |
| annual minima of 7-day mean river flow | G2G_GB_aminflow_obs_1980_2011.nc<br>G2G_NI_aminflow_obs_1980_2011.nc | Dec 1980–Nov 2011 |
| monthly mean soil moisture content | G2G_GB_mmsoil_obs_1980_2011.nc<br>G2G_NI_mmsoil_obs_1980_2011.nc | Dec 1980–Nov 2011 |


**Table 3 The file naming convention for the climate projection-based datasets.**

| Data | Names of NetCDF files | Years available |
|------|----------------------|-----------------|
| monthly mean river flow | G2G_GB_mmflow_UKCP18RCM_ensnum_1980_2080.nc<br>G2G_NI_mmflow_UKCP18RCM_ensnum_1980_2080.nc | Dec 1980–Nov 2080 |
| annual maxima of daily mean river flow, and dates of occurrence | G2G_GB_amaxflow_UKCP18RCM_ensnum_1980_2080.nc<br>G2G_NI_amaxflow_UKCP18RCM_ensnum_1980_2080.nc | Oct 1981–Sep 2080 |
| annual minima of 7-day mean river flow, and dates of occurrence | G2G_GB_aminflow_UKCP18RCM_ensnum_1980_2080.nc<br>G2G_NI_aminflow_UKCP18RCM_ensnum_1980_2080.nc | Dec 1980–Nov 2080 |
| monthly mean soil moisture content | G2G_GB_mmsoil_UKCP18RCM_ensnum_1980_2080.nc<br>G2G_NI_mmsoil_UKCP18RCM_ensnum_1980_2080.nc | Dec 1980–Nov 2080 |

ensnum is the number of the ensemble member (01, 04, 05, 06, 07, 08, 09, 10, 11, 12, 13, 15)


For the observation-based datasets, the time stamp in the NetCDF files is "days
since 1900-01-01", and the monthly mean river flows and monthly mean soil
moisture are nominally assigned to the first day of the month. The annual
maximum/minimum flow values are nominally assigned to the start year of the 12-
month period over which they are calculated, e.g. the annual maximum flow
assigned to 1981 is for 1/10/1981–30/9/1982 (water years), while the annual
minimum flow assigned to 1981 is for 1/12/1981–30/11/1982 (Dec–Nov years). The
'time_bnds' variable gives the start and end dates of the time period over which the
annual maximum or minimum flow are extracted.
For the climate projection-based datasets, the data have 30-day months due to the
"360_day" calendar of the Hadley Centre climate model. The files are otherwise as
above, except that the dates of occurrence of the annual maximum and minimum
flows are also provided, as additional variables in the 'amaxflow' and 'aminflow'
NetCDF files respectively.
Table 4 summarises the spatial domains covered by the GB and NI datasets. River
flows are only provided for non-sea and non-tidal river cells with a catchment area of
at least 50km$^2$, and set to missing elsewhere. Soil moisture estimates are provided
for all non-sea cells, and set to missing elsewhere.

**Table 4 Summary of domain sizes and extents, including the OSGB co-ordinates for**
**the lower left and upper right corners (m).**

|  | GB | NI |
|---|---|---|
| Domain size | 700 km × 1000 km | 187 km × 170 km |
| Lower left corner | (0,0) | (-7000,440000) |
| Upper right corner | (700000,1000000) | (180000,610000) |


To aid use of the datasets, further data files are provided for both GB and NI,
including catchment area grids, grids identifying majority lake cells, and grids
identifying the approximate locations of river flow gauging stations (Table 5). The
catchment area grids are mapped in Figure 1, while the majority lake cells and
gauging station locations are mapped in Figure 2 (note that although GB and NI are
mapped together, the data for GB and NI are provided separately). At the gauging
station locations the G2G flow estimates can be compared to observed river flows.

**Table 5 The additional data files for GB and NI.**

| Data | File names | Description |
|---|---|---|
| Catchment area grid | UKSCAPE_G2G_GB_CatchmentAreaGrid.nc<br>UKSCAPE_G2G_NI_CatchmentAreaGrid.nc | Digitally-derived catchment area ($km^2$) draining to every 1km x 1km grid box. |
| Majority lake cells grid | UKSCAPE_G2G_GB_SoilMoisture_LakeGrid.nc<br>UKSCAPE_G2G_NI_SoilMoisture_LakeGrid.nc | Cells with greater than 85% of area covered by water (according to 25m data from Land Cover Map 2015, Rowland et al. 2017a,b). These grids can be applied to exclude use of soil moisture data in majority lake cells. 1=land, 2=lake, and -9999=sea. |
| | UKSCAPE_G2G_NI_LakeGrid.nc | As above but cells with greater than 70% of area covered by water, plus some manual additions of cells for Lough Erne to avoid more than one change from river to lake to river for each flow pathway. This grid can be applied to exclude use of river flow data in lake cells in NI. |
| Gauging station location grid | UKSCAPE_G2G_GB_NRFAStationIDGrid.nc<br>UKSCAPE_G2G_NI_NRFAStationIDGrid.nc | Best locations corresponding to 1038 gauging stations in GB and 43 gauging stations in NI, referenced by NRFA station number (nrfa.ceh.ac.uk). NRFA station number at gauging station locations, 0=land, and -9999=sea. |
| Gauging station info | UKSCAPE_GB_NRFAStationIDs.csv<br>UKSCAPE_NI_NRFAStationIDs.csv | Information on stations included in location grids. Information for 18 additional stations is included in the GB file; these are each located in the same 1km cell as one of the stations in the grid (as detailed in the comments column of the csv file). |



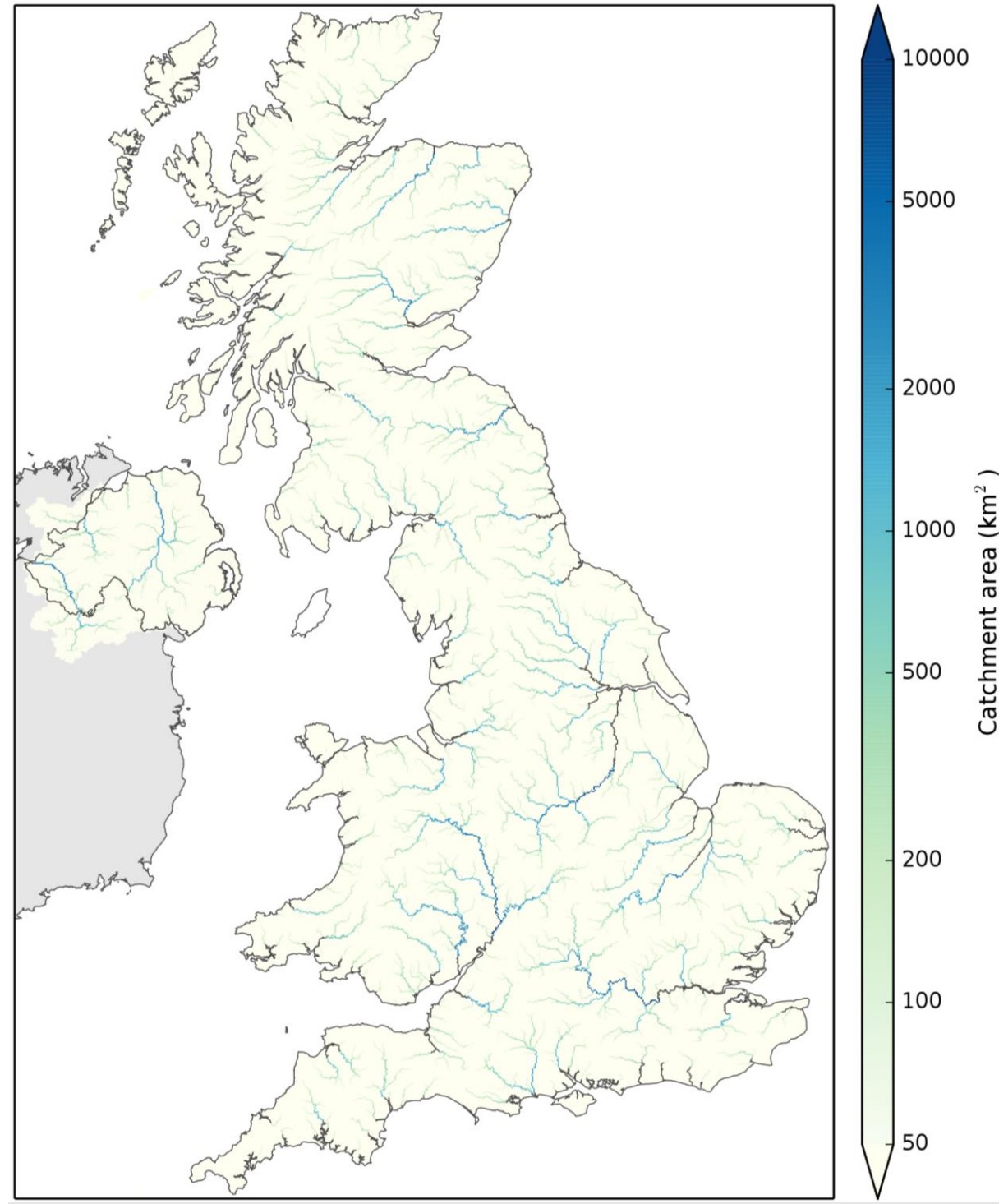

**Figure 1 Map showing the catchment area grids for GB and NI (see Table 5).**


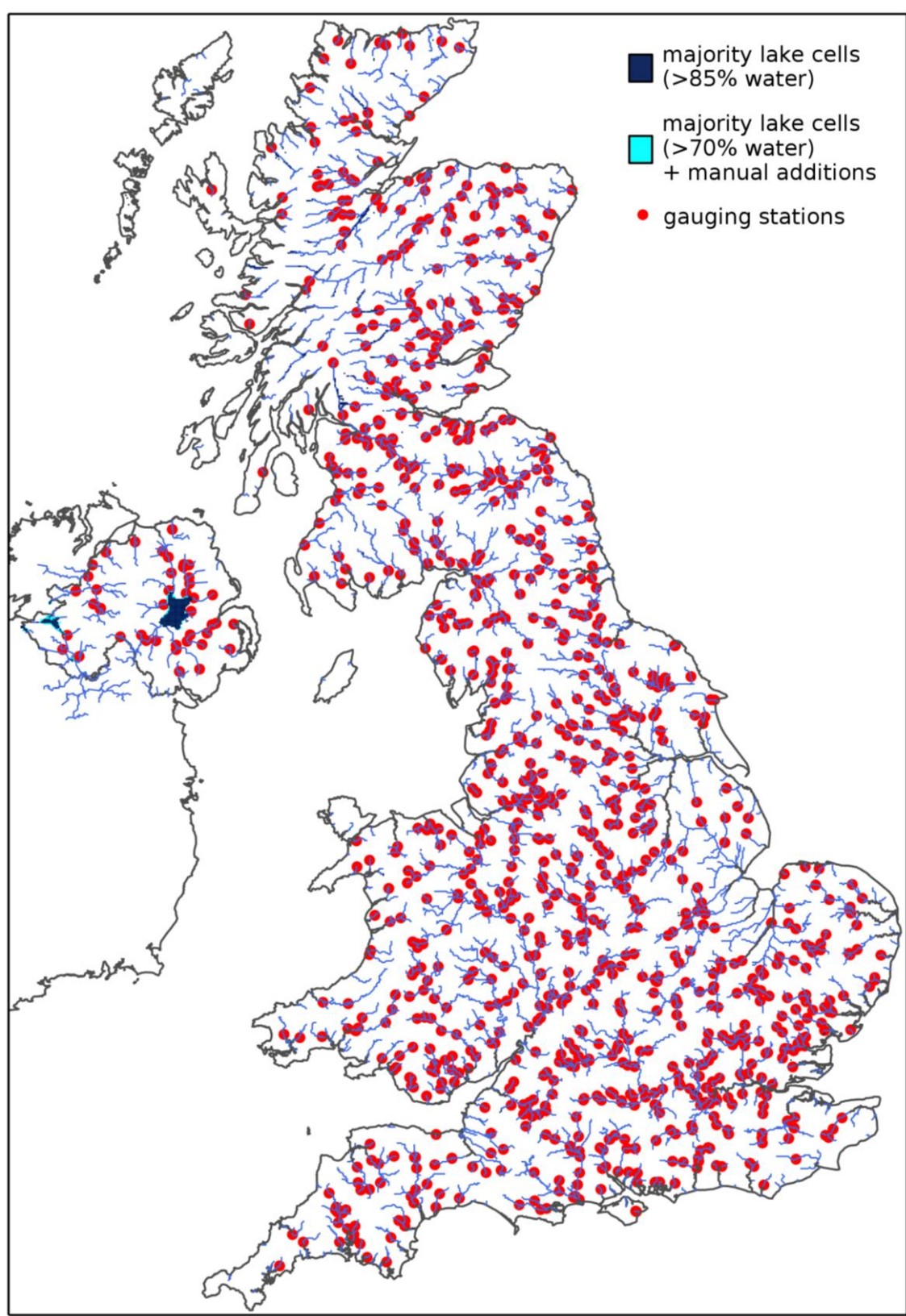


**Figure 2 Map showing the majority lake cells and gauging station locations for GB and NI (see Table 5), along with the main rivers (catchment area ≥ 50km$^2$; blue lines). Note that the 'majority lake cells (>70% water)' cover small areas around Lough Neagh and Lough Erne in Northern Ireland only.**

## 2.6  How to use the datasets

River flows from the observation-based simulation can be compared to observed (gauged) river flows (for example from the National River Flow Archive, NRFA; nrfa.ceh.ac.uk), and to facilitate such comparisons files identifying gauging station locations on the 1km G2G model grid for GB and NI are provided (see Table 5). However, it should be borne in mind that G2G provides natural flow estimates, so comparisons in catchments affected by artificial influences like abstractions and discharges may not be as good as those in catchments with relatively natural flow regimes (Rameshwaran et al. 2022). Also, although the gauging station locations have been identified as the G2G cell closest in terms of geographical location and catchment area, and checks have been undertaken to ensure that the G2G flows are for the correct river rather than a nearby river with a similar catchment area, in some cases the derived catchment area draining to the 1km x 1km cell will be different to the "observed" NRFA catchment area. This problem can particularly affect smaller catchments, for which discretisation to a 1km x 1km grid can lead to proportionally larger errors, although flow data provided here are in any case limited to catchments with drainage areas of at least $50km^2$. The catchment area grids (Table 5) can be used to check the drainage area of gauged catchments, and could also be used to identify the most appropriate 1km x 1km cell corresponding to any particular ungauged catchment of interest.

Users should be aware that the effect of water bodies such as lakes and reservoirs is not accounted for within the model; any impact of lake storage and regulation on downstream river flows has been neglected, and lake grid-cells are treated as though they were rivers. The data files thus include 'river flows' and 'soil moisture' for 1km cells located within lakes. Additional files identify majority lake cells in GB and NI, so that these can be excluded from analyses if desired (see Table 5).

The historical portion of the climate projection-based river flow and soil moisture datasets can be compared to the observation-based datasets, or to observed data. However, comparisons in either case should only be made statistically (over multi-decadal periods), not directly (time point by time point), as there will be no equivalence between observed weather features and those in the RCM PPE at the same date. An example of such a comparison is presented in Supp. Fig. 4 of Kay et al. 2021a, where mean monthly flows, flood frequency curves and low flow frequency curves are compared for the baseline SIMRCM ensemble and SIMOBS, for 8 catchments in NI. Comparison of climate projection-based simulations to the observation-based simulation will indicate how both natural variability and (remaining) biases in the climate projection data affect the hydrological model simulations for the baseline period, while comparison to observed data themselves will be additionally affected by the accuracy of the G2G model.

The climate projection-based datasets for baseline (historical) and future periods can be compared statistically, to investigate the potential future impacts of climate change on river flows (e.g. Kay 2021, Lane & Kay 2021, Kay et al.2021a) and soil moisture (e.g. Kay et al. 2022a). Analyses should use the full ensemble; each member should be considered as a different but plausible realisation. Comparison between periods should use the same ensemble member for each period, not a baseline from one member and a future from another member.

The observation-based datasets for GB can be considered updates to MaRIUS-G2G-MORECS-monthly flow and soil moisture data (Bell et al. 2018a). The main

differences are the shorter simulation period here (Dec 1980–Nov 2011 vs 1960–
2015 or 1891–2015), the inclusion here of the optional snow module, some changes
to the land-sea mask, some changes related to infilling of missing soil type data, and
minor changes to the discretised river flow network to improve the G2G model
catchment areas (thus the additional spatial datasets provided here may differ in
places to those provided with the MaRIUS dataset).
The climate projection-based datasets for GB are analogous to the MaRIUS-G2G-
WAH2-monthly flow and soil moisture data (Bell et al. 2018b), which were driven by
weather@home climate model data (Guillod et al. 2017). The main differences, as
well as the factors listed above for the observation-based datasets, are the climate
model version, the smaller ensemble size here (12 members vs 100 members), and
the provision here of transient rather than time-slice data (Dec 1980–Nov 2080 vs
1900–2006, 2020–2049 and 2070–2099). Note that, as far as the authors are aware,
there has been no comparison between the weather@home and UKCP18 RCM
climate datasets.

# 3  Results

## 3.1  Monthly mean river flows

Maps of example monthly mean river flows across GB and NI from SIMOBS and two
SIMRCM ensemble members (Figure 3) illustrate the accumulation of water as it
flows downstream, with typically higher flows for downstream locations with larger
catchment areas. The example maps also show the generally lower flows in summer
(July) compared to winter (January). Note that, although GB and NI are mapped
together, the data for GB and NI are provided separately.

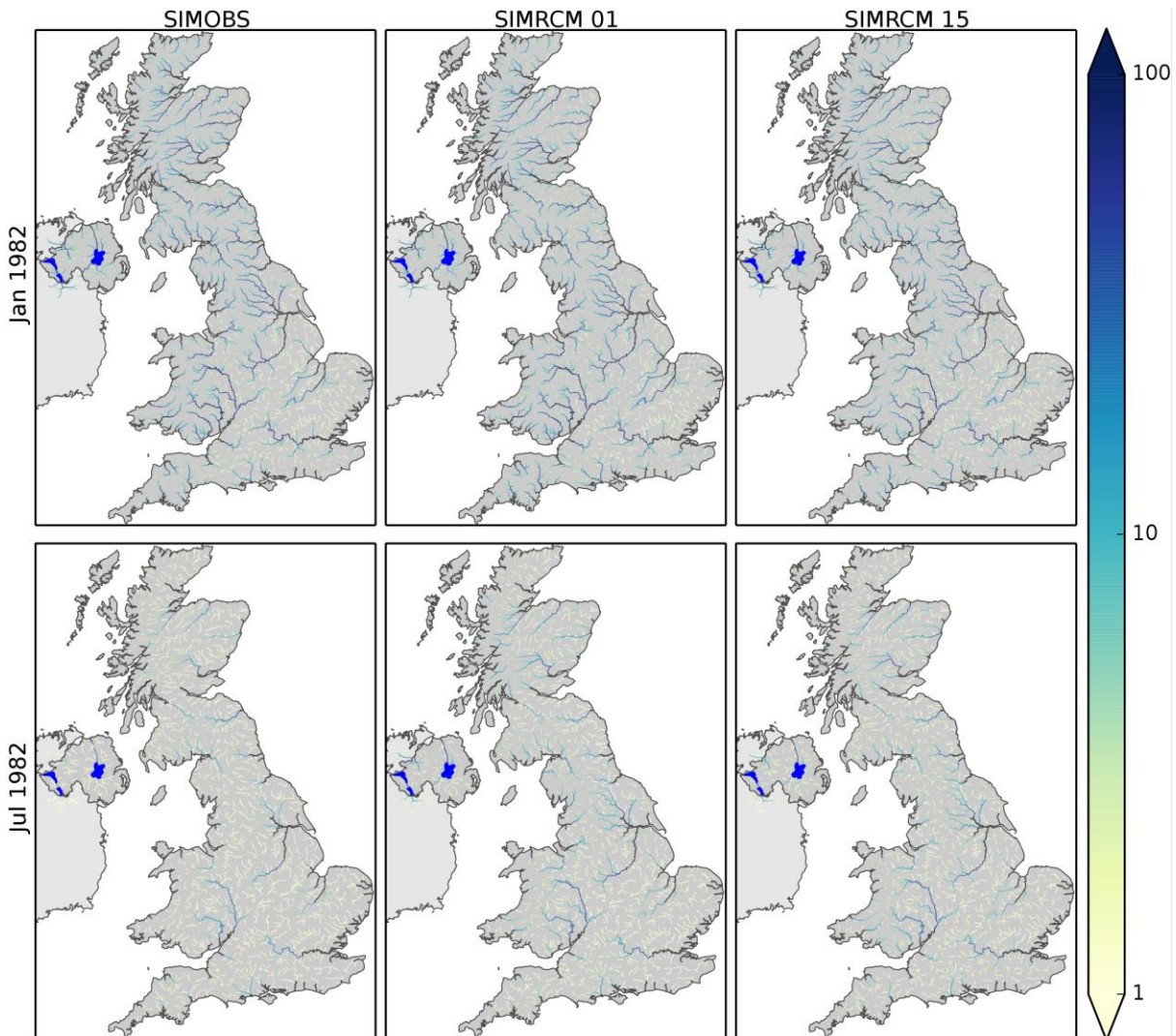


**Figure 3 Maps of monthly mean river flows (m³s⁻¹) for January and July 1982, from SIMOBS (left) and two SIMRCM ensemble members (01 – centre, and 15 – right). Also shown are Lough Neagh and Lough Erne in NI (bright blue shading).**


Time-series plots of UK-mean annual mean river flows from SIMOBS and the SIMRCM ensemble show good correspondence (Figure 4). There is a relatively small but highly statistically significant decrease in the SIMRCM ensemble mean flow over Dec 1980–Nov 2080 (-0.695 m³s⁻¹ / 100 years) (Figure 4). Six of the 12 individual ensemble members show decreases significant at the 10% level, while four show non-significant decreases and two show non-significant increases. Plots of the monthly climatology of UK-mean river flows for the first and last 30 years (Dec 1980–Nov 2010 and Dec 2050–Nov 2080) show a clear reduction in flows during summer and early autumn, but a possible increase in winter (Figure 4).

308

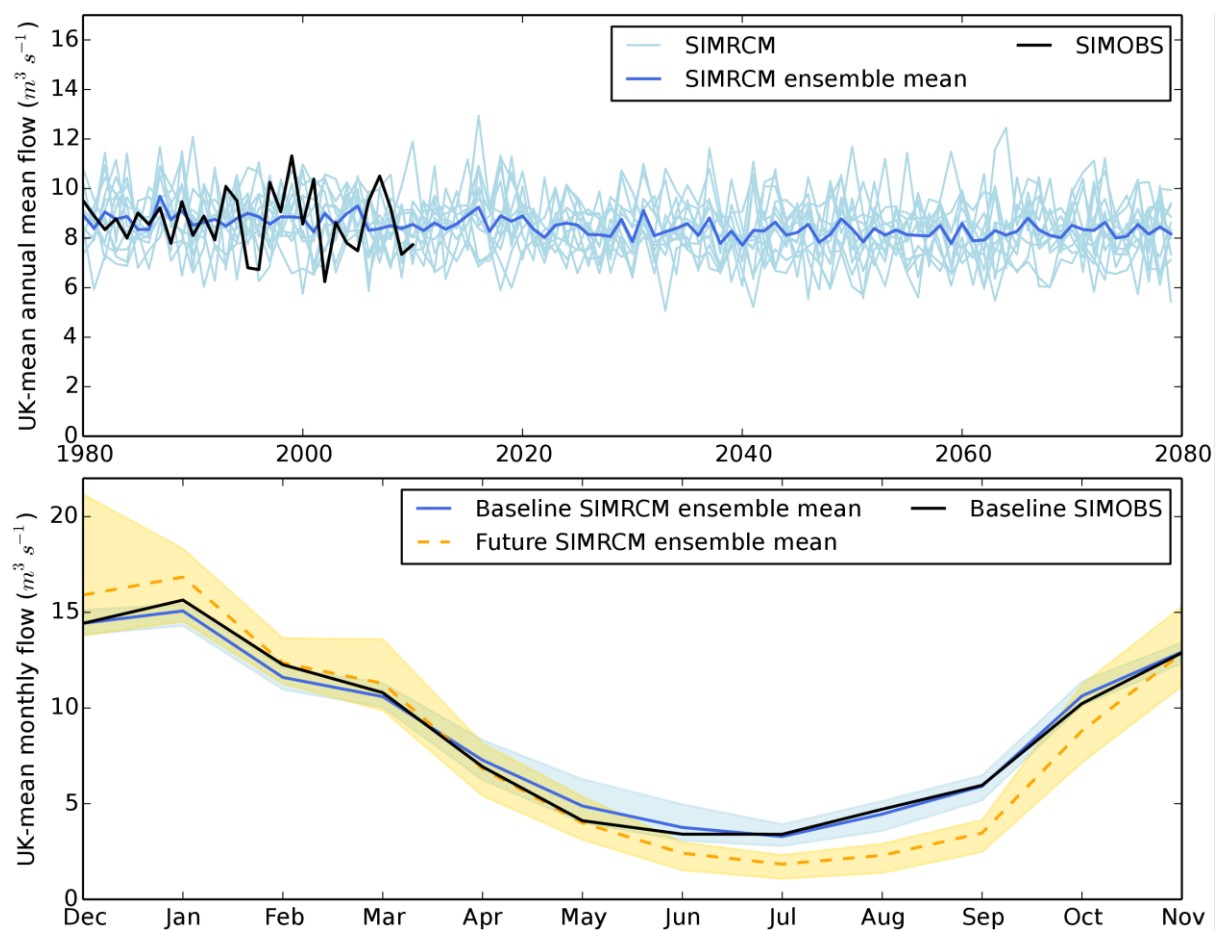

**Figure 4 Time-series of UK-mean annual mean river flows (top), and the baseline (Dec 1980–Nov 2010) and future (Dec 2050–Nov 2080) monthly climatology of UK-mean river flows (bottom), for SIMOBS and the SIMRCM ensemble. The shading in the bottom plot shows the SIMRCM ensemble range for each period.**

Kay (2021) used the GB SIMRCM monthly mean river flow data to investigate potential future changes in seasonal mean flows, for two future time-slices (2020–2050 and 2050–2080). This suggested large decreases in summer mean flows everywhere, but possible increases in winter mean flows, especially in the north and west. A similar analysis using the NI SIMRCM monthly mean river flow data (Kay et al. 2021a) suggested decreases in spring–autumn mean flows, especially in summer, but possible increases in winter mean flows.

## 3.2   Extreme river flows

Maps of example GB and NI AMAX of daily mean river flows and AMIN of 7-day mean river flows from SIMOBS and two SIMRCM ensemble members (Figure 5) show less spatial variation than those of monthly mean river flows (when plotted on the same scale). Note that, although GB and NI are mapped together, the data for GB and NI are provided separately.

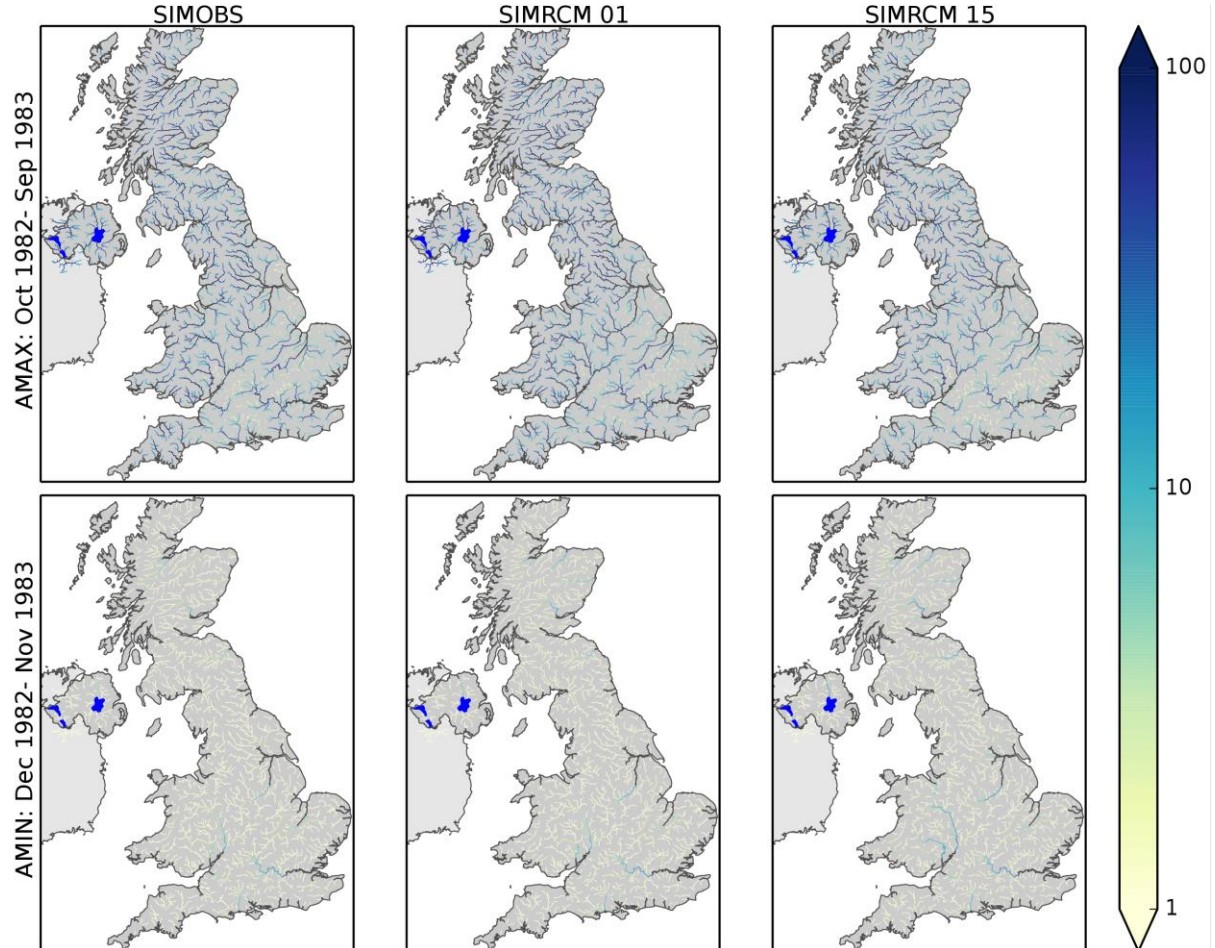

329

**Figure 5 Maps of AMAX of daily mean river flows for Oct 1982–Sep 1983 (m³s⁻¹; top)**
**and AMIN of 7-day mean river flows for Dec 1982–Nov 1983 (m³s⁻¹; bottom), from**
**SIMOBS (left) and two SIMRCM ensemble members (01 – centre, and 15 – right). Also**
**shown are Lough Neagh and Lough Erne in NI (bright blue shading).**

334

Time-series plots of UK-mean AMAX and AMIN river flows from SIMOBS and the
SIMRCM ensemble show good correspondence (Figure 6). The SIMRCM ensemble
mean AMAX flows show a statistically significant increase over Oct 1981–Sep 2080
(8.51 m³s⁻¹ / 100 years) (Figure 6). Nine of the 12 individual ensemble members
show increases in AMAX flows significant at the 10% level, while one shows non-
significant increases and two show non-significant decreases. The SIMRCM
ensemble mean AMIN flows show a highly statistically significant decrease over Dec
1980–Nov 2080 (-0.670 m³s⁻¹ / 100 years) (Figure 6), and all 12 individual ensemble
members show decreases significant at the 10% level.

344

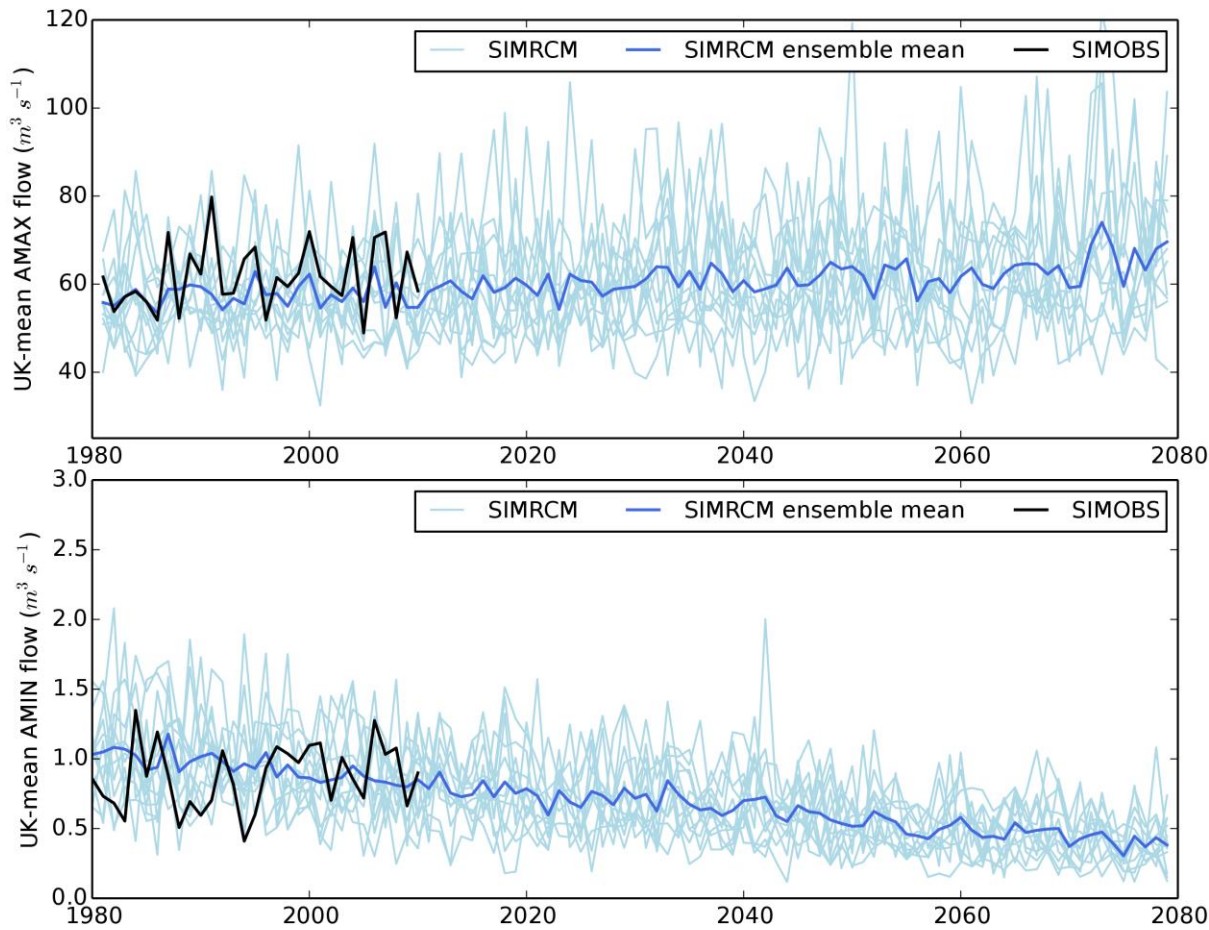

**Figure 6 Time-series of UK-mean AMAX of daily mean river flows (top) and AMIN of 7-day mean river flows (bottom), for SIMOBS and the SIMRCM ensemble.**

Lane & Kay (2021) used the GB SIMRCM AMAX and AMIN river flow data to investigate potential future changes in high/low flows by 2050–2080. All ensemble members showed large reductions in 10-year return period low flows. The direction of change for 10-year return period high flows was more uncertain, but increases of over 9% were possible in most areas. Simultaneous worsening of both high and low flow extremes was projected in the west. A similar analysis using the NI SIMRCM AMAX and AMIN flow data (Kay et al. 2021a) suggested large reductions in 10-year return period low flows everywhere, and large increases in 10-year return period high flows for some locations and ensemble members. Analyses of the GB and NI dates of occurrence of SIMRCM AMAX and AMIN showed few significant changes in timing (Lane & Kay 2021, Kay et al. 2021a).

## 3.3   Soil moisture

Maps of example GB and NI monthly mean soil moisture content from SIMOBS and two SIMRCM ensemble members (Figure 7) show the spatial variation, which is generally related to the variation in soil types. The example maps also show the generally drier soils in summer (July) compared to winter (January), and show differences between the two selected SIMRCM ensemble members. Note that,

although GB and NI are mapped together, the data for GB and NI are provided
separately.

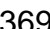

**Figure 7 Maps of monthly mean soil moisture content (m water / m soil) for January**
**and July 1982, from SIMOBS (left) and two SIMRCM ensemble members (01 – centre,**
**and 15 – right).**

Time-series plots of UK-mean annual mean soil moisture content from SIMOBS and
the SIMRCM ensemble show good correspondence (Figure 8). The SIMRCM
ensemble mean soil moisture content shows a highly statistically significant
decrease over Dec 1980–Nov 2080 (-0.035 / 100 years) (Figure 8), and all 12
individual ensemble members show decreases significant at the 10% level. Plots of
the monthly climatology of UK-mean soil moisture content for the first and last 30
years (Dec 1980–Nov 2010 and Dec 2050–Nov 2080) show a clear reduction in
summer and autumn (Figure 8).

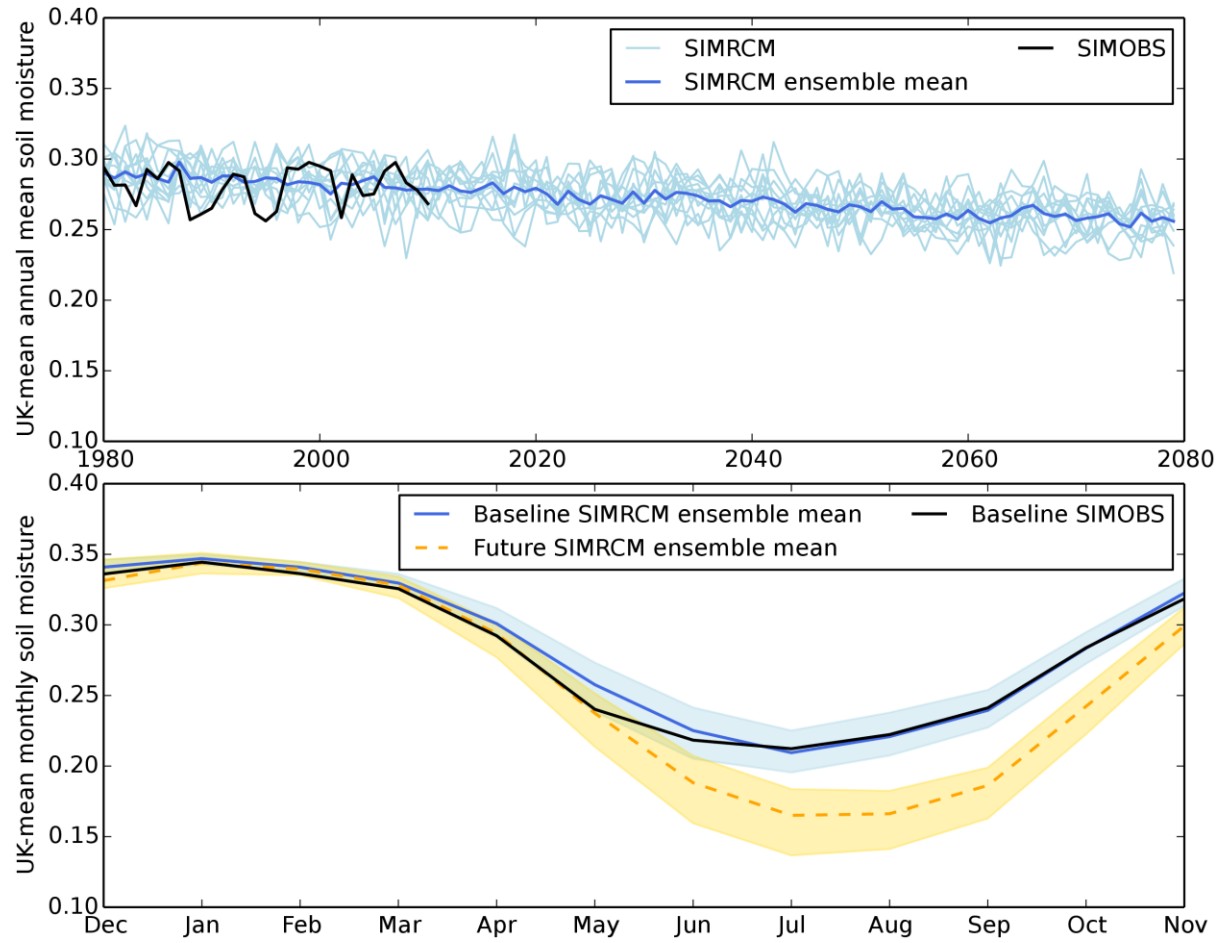

**Figure 8 Time-series of UK-mean annual mean soil moisture content (m water / m soil; top), and the baseline (Dec 1980–Nov 2010) and future (Dec 2050–Nov 2080) monthly climatology of UK-mean soil moisture content (m water / m soil; bottom), for SIMOBS and the SIMRCM ensemble. The shading in the bottom plot shows the SIMRCM ensemble range for each period.**

Kay et al. (2022a) used the GB and NI SIMRCM monthly mean soil moisture data to investigate potential future changes in occurrence of indicative soil moisture extremes and changes in typical wetting and drying dates of soils by 2050–2080 across the UK. This also suggested large increases in the spatial occurrence of low soil moisture levels, and later soil wetting dates. Changes to soil drying dates were less apparent.

# 4  Discussion

Ensemble data from the historical period of the climate projection-driven datasets show good correspondence with the observation-driven datasets, for both river flows (Figure 4 and Figure 6) and soil moisture (Figure 8). More detailed performance analyses are provided elsewhere (Kay 2021, Lane & Kay 2021, Kay et al. 2021a, Kay et al. 2022a).

The climate projection-driven river flow and soil moisture datasets suggest potential future decreases in summer flows, annual minimum 7-day flows, and summer/autumn soil moisture, along with possible future increases in winter flows and annual maximum flows. More detailed analyses, presented elsewhere, illustrate

the variation in these general trends, both spatially and by ensemble member (Kay 2021, Lane & Kay 2021, Kay et al. 2021a, Kay et al. 2022a). These changes are consistent with the climate projections, which give wetter winters and drier and hotter summers (Murphy et al. Fig. 5.2), and increased summer PE (Robinson et al. 2022).

A study of trends in historical gauged flows from the UK Benchmark Network (Harrigan et al. 2017) shows a tendency for an increase in winter mean flows and high flow indices over the past 50 years, although with significant natural decadal variability (clear so-called flood-rich and flood-poor periods). The datasets here suggest that this overall trend could continue into the future, and they could potentially be used to further investigate natural variability. The analysis of Harrigan et al. (2017) shows less consistent changes in summer mean flows or low flow indices, with catchments in the south/east often showing decreases, but catchments in the north/west typically showing increases. The datasets here suggest that more consistent decreases could be seen everywhere in future.

However, the fact that the UKCP18 Regional climate projections applied here only use one GCM/RCM needs to be borne in mind. Other climate models tend to give smaller decreases (or increases) in summer precipitation than the UKCP18 Regional projections (Murphy et al. 2018 Fig. 5.2), so are likely to give lower reductions in summer flows and soil moisture. Similarly, other climate models give a wider range of changes in winter precipitation than the UKCP18 Regional projections (Murphy et al. 2018 Fig. 5.2), so could give larger or smaller increases in winter flows. The UKCP Local projections, produced by nesting a ~2.2km convection-permitting model in each RCM PPE member (Kendon et al. 2021), also give some differences in climatic changes compared to the RCM (Kendon et al. 2021), and consequently some differences in hydrological impacts (Kay 2022). In addition, the use of a high emissions scenario (RCP8.5) for the UKCP18 Regional projections is likely to lead to more extreme changes than would occur for lower emissions (e.g. Arnell et al. 2014), although RCP8.5 should not be considered implausible (Schwalm et al. 2020).

Further sources of uncertainty in the datasets include the observation-based PE and the calculation of RCM PE. MORECS 40km monthly PE data are used for the observation-based hydrological model runs; lower spatial and temporal resolution than the other driving data (Section 2.2). A dataset of 1km daily PE has since been derived (Brown et al. 2022) using HadUK-Grid data (Met Office 2019), although some the variables required had to be interpolated from monthly to daily. The RCM PE used here includes the effect of stomatal closure under higher $CO_2$ concentrations but does not include a potential leaf area increase due to carbon fertilisation (Rudd and Kay 2016; Robinson et al.2022).

Potential future changes in land cover are also excluded, as are any artificial influences on river flows. Also, only one hydrological model has been applied; a catchment-based dataset of simulated river flows from the 'Enhanced future Flows and Groundwater' (eFlaG) project (Hannaford et al. 2022b) uses similar driving data from the UKCP18 Regional projections for three hydrological models (including G2G), so could be used to look at potential uncertainty from hydrological model structure (Hannaford et al. 2022a). Note that eFlaG used HadUK-Grid rainfall, both for observation-based runs and for bias correction of UKCP18 RCM data, whereas CEH-GEAR rainfall were used here since data are included for the parts of the Republic of Ireland draining into Northern Ireland, enabling gridded flow simulation across Northern Ireland. The ability to provide full and coherent coverage, of gauged

and ungauged locations is a particular strength of national-scale grid-based models
like G2G, in contrast to catchment-based models.

# 5  Conclusions

The datasets presented here provide consistent spatial simulations of river flows and
soil moisture for the whole of the UK, driven by both observed data and by an
ensemble of regional climate model data from the latest UK climate projections,
UKCP18. These enable direct studies of historical and potential future river flows and
soil moisture, but they can also be used to provide inputs for further studies, for
example to simulate water quality (e.g. Hutchins et al. 2016), crop yields (e.g. Cai et
al. 2009), irrigated agriculture economic risk (Salmoral et al. 2019), or ecological
impacts (e.g. Bussi et al. 2016).
An online (anonymous) stakeholder survey was carried out for UK-SCAPE WP2.2 in
late 2021 (Kay et al. 2022e). This asked a set of questions divided into three broad
classes; 'job role and level of experience', 'data of interest', and 'data format/access
preferences'. The responses on 'data of interest' showed that there is a lot of interest
in water quantity, including both river flows and soil moisture, and a lot of interest in
potential future changes in river flows, although slightly less so for changes in soil
moisture. Furthermore, the responses on 'data format/access preferences' showed
that the greatest 1$^{st}$ preference was for grids covering sub-regions or the whole
country, although perhaps unsurprising this varied by job role (Academic,
Government/Regulator, Practitioner/ Consultant), which likely influences how data
are used. A large proportion of respondents were also happy downloading the full
dataset as NetCDF files from the EIDC. The datasets described here thus provide for
a significant stakeholder demand, although there is always more that could be done.
Further developments could include a web-tool allowing interactive data exploration
and plotting.

# Acknowledgements

This work was supported by the Natural Environment Research Council award
number NE/R016429/1 as part of the UK-SCAPE programme delivering National
Capability. Thanks to Emma Robinson (UKCEH) for work on the estimation of PE
from climate model data.

# Data Availability

The six datasets described in this manuscript are available from the Environmental
Information Data Centre (EIDC); see Table 1.

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
