# Peer review of "The UKSCAPE-G2G river flow and soil moisture datasets: Grid-to-Grid model estimates for the UK for historical and potential future climates"

_Earth System Science Data, 2022_

## Author Comment (AC1)

**Response to reviewers' comments on "The UKSCAPE-G2G river flow and soil moisture datasets: Grid-to-Grid model estimates for the UK for historical and potential future climates" by Kay et al.**

**Reviewer 1:**
This data paper serves as a nice bringing together of a variety of existing, but closely related datasets around the G2G national modelling. I find that having all of these details together in one place will be useful for a wide variety of UK-based research and modelling. I have tested all data links for the datasets presented and they currently work (2023-03-10). I have found the paper to be well written with only some minor comments below.
I would note that I have selected good rather than excellent for originality and uniqueness because this paper is bringing together existing data rather than presenting new data. Though as above, I think this is still useful to bring these datasets under the same lens.
Thank you. We describe below our response to each comment.

Minor comments:
I haven't reviewed an ESS data paper before, but it seems odd to me to have what appears to be acknowledgements as the first paragraph of an introduction. Maybe for this paper format it is fine.
Given the title of the paper, we felt it was useful to mention the UKSCAPE programme early on. We would be happy to change this at the editor's request.

L98 - The authors may wish to note that the HadUK data used provides all the necessary variables to calculate PE if users require a higher resolution representation of PE. I understand that the intention of this data paper is to draw reader's attention to the soil moisture and flow datasets produced, but I think this addition would be useful.
The possible future use of PE derived from HadUK-Grid (Brown et al. 2022) will be added (Section 4 para' 4), although unfortunately many of the required variables are only available at a monthly time-step so have to be interpolated to daily anyway.
Brown, M.J. et al. (2022). Potential evapotranspiration derived from HadUK-Grid 1km gridded climate observations 1969-2021 (Hydro-PE HadUK-Grid). NERC EDS Environmental Information Data Centre. doi:10.5285/9275ab7e-6e93-42bc-8e72-59c98d409deb.

L109 - I am missing some mention about why the convection permitting simulations weren't used. It seems this would overcome some of the steps needed in S2.3 and be more accurate in general. At least readers should be made aware of its existence.
The UKCP Local CPM-based dataset was not available at the time of the work reported in this manuscript, but some comparisons have since been done and reference to this will be added (Section 4 para' 4).

Figure 2 - the only lake cells I can identify are the 2 Northern Irish lakes. Are we supposed to be able to spot more? If so they will need more highlighting - or more explicit linking to the text about how the lakes are mainly significant in NI. Otherwise I'm not quite sure the point of Fig 2.
There are some other lakes mapped, particularly in Scotland, which can be seen if you zoom in on the map in Figure 2 (note that the final version of the paper will contain higher resolution maps than provided for review). Section 2.1 para' 2 explains that the effect of lakes is minimal in GB (largest lake in Scotland ~71km$^2$ and largest in England ~15km$^2$) but more important in NI (Lough Neagh ~390km$^2$ and Lough Erne ~144km$^2$). Figure 2 also shows the gauging station locations.

L238 - I think this paragraph can be written more clearly. Maybe it should start with "For the historical portion of the RCM PPE projections,.. "? But if so, it seems to overlap with the use of 'baseline periods' in the following paragraph (starting L251). I'm still a bit confused by it.

The start of the paragraph will be changed to read "The historical portion of the climate projection-based river flow and soil moisture datasets can be compared to the observation-based datasets…", and the start of the following paragraph will be edited to make the terminology more consistent, by stating "...baseline (historical)…".

L267 - If my passing understanding of w@h and UKCP18 is correct (which it may not be), these two datasets can result in different (significantly different?) distributions of (e.g.,) precipitation, particularly at extremes. If this is true, it should be mentioned here. If there is no study that has made this comparison, then that is important information too.
It is entirely possible that the w@h and UKCP18 data are different, especially for extremes given the much larger ensemble size of the former compared to the latter, but we're not aware of a specific comparison of them – this will be clarified (Section 2.6 para' 6).

Section 3.3 - It seems that the AMIN/AMAX soil moisture figure is missing here. It may not be such a conventionally studied metric but I feel is important to highlight the extremes.
Unfortunately, only monthly mean soil moisture grids were produced, not annual minima or maxima (Table 2). We will bear your suggestion in mind for future datasets.

L416 - The authors may find it helpful to cite Schwalm et al. (2020) for this statement and thus support their choice of rcp8.5. Also if I remember rightly there is only RCP8.5 for the UKCP18 regional projections anyway, which may also be worth mentioning and further justifies the use of this RCP.
The reference will be added (Section 4 para' 4), and the fact that the Regional projections are only available for RCP8.5 will be emphasised (Section 2.3 para' 1).

Editorial:
L40 & L215 - hyperlinks don't seem to work
This might just be a problem in the version provided for review; they work fine in our version.

L324 - 'highly statistically significant increase' -> 'statistically significant increase'
Will be changed.

---

## Author Comment (AC2)

**Response to reviewers' comments on "The UKSCAPE-G2G river flow and soil moisture datasets: Grid-to-Grid model estimates for the UK for historical and potential future climates" by Kay et al.**

**Reviewer 2:**
The paper presents several datasets of UK river flows, soil moisture and derived statistics using both historical (CEH-GEAR precipitation, MORECS PET, HadUK-Grid min/max Temperature) and UKCP18 12km regional projection climate data. The hydrological model used for producing the datasets is the G2G model set up at a 1x1km spatial resolution and 15-min temporal resolution. The paper is generally well written. I only have a few technical comments as follows.
Thank you. We describe below our response to each comment.

The river flow dataset provided is monthly mean, but the G2G model operates at a 15-min time step. Please explain why the daily river flow dataset is not provided. The daily flows can be more useful for the users.
While it is possible to produce 1km gridded daily time-series from G2G, these are not typically produced as they are very large files (especially if long time periods are covered, as in this case). Instead the annual maxima and minima are calculated and saved during the model run, to enable analyses of high and low flows without having to save the daily gridded flows. This will be clarified (Section 2.4).

The authors cited the performance of the G2G model in Lines 77-79. But it is not straightforward to find the performance of the catchments included in the datasets provided in this paper, because the GB and NI catchments were presented in separate papers and it is also not clear which catchments were included in this paper. I feel it is necessary to include the model performance of the selected catchments in this paper along with the datasets in a table format. This information should be added to Table 5.
As we do not provide daily flow time-series for specific catchments, we also don't provide information on performance for specific catchments. That form of analysis has been done previously, although not for all of the catchments whose locations are given in the NRFAStationIDGrid files – for example, some have insufficient gauged flows available in the period. We provide the NRFA catchment locations purely to make it easier for a user to sub-select data corresponding to a specific catchment or catchments if that is what they require, in which case they could also do an assessment of performance in a way that is directly relevant to their application (as stated at the end of Section 2.5). Equally, flows can be selected for ungauged locations of interest, or gridded time-series can be used. Clearly, no performance assessment is possible for ungauged locations. The full and coherent coverage of gauged and ungauged locations is a particular strength of models such as G2G, in contrast to outputs from catchment-based models.

Line 174-177: The annual maximum/minimum flow values are nominally assigned to the start year of the 12-month period over which they are calculated, e.g. the annual maximum flow assigned to 1981 is for 1/10/1981–30/9/1982 (water years), while the annual minimum flow assigned to 1981 is for 1/12/1981–30/11/1982 (Dec–Nov years). It is reasonable to use the two different water years here, but it is necessary to explain to the reader why the difference and if it affects the statistics of maxima and minima derived from the daily flows.
The AMAX of daily mean flows are extracted for water years (Oct–Sep), to try to avoid extraction of the same high flow event in consecutive years. AMIN extraction would usually use calendar years, but Dec-Nov is used here to match the climate model data

running from December 1980 to November 2080, whilst still trying to avoid extraction of the same low flow event in consecutive years. This will be clarified (Section 2.4).

Please explain if the dataset of river flows is similar as the one presented in the other paper https://doi.org/10.5194/essd-2022-40 using the same G2G model. The latter contains daily flows from three models but only at the catchment outlets. Are the forcing inputs the same?
The driving data applied here are not exactly the same as for the eFlaG dataset. The eFlaG project used HadUK-Grid rainfall for their observation-based runs, whereas here we used CEH-GEAR rainfall to enable simulation of river flows across Northern Ireland (for which data covering some parts of the Republic of Ireland are required, which are available from CEH-GEAR but not from HadUK-Grid). Because of this, the bias correction of the UKCP18 RCM rainfall data was performed against HadUK-Grid in eFlaG but against CEH-GEAR here – the correction grids are similar but not exactly the same, so the SIMRCM runs will not be exactly the same. This will be clarified (end of Section 4).

Figure 1 does not present much useful information as one cannot really tell the catchment area grids from the colours. I am unsure if this Figure should remain in the main paper.
We prefer to keep Figure 1 as we believe that it helps a user to visualise how the grid-based modelling works. It also clearly shows the parts of the Republic of Ireland needed for simulating river flows in Northern Ireland.

Figure 2 is supposed to show majority lake cells >85% and >70%. I can clearly see the large lake (Lough Neagh) >85% but not sure where the area of lake cells>70% is. It might be the Lough Erne but not clear. It needs to be either noted in the text or highlighted in the map.
The 'lake cells>70%' cover small areas around both Lough Neagh and Lough Erne, which can be seen if you zoom in on the map in Figure 2 (note that the final version of the paper will contain higher resolution maps than provided for review). This will be clarified in the caption.

---

## Author Response (AR2)

**Response to Editor's comments on "The UKSCAPE-G2G river flow and soil moisture datasets: Grid-to-Grid model estimates for the UK for historical and potential future climates" by Kay et al.**

Thank you for properly addressing the reviewers' comments in your revised manuscript. I am happy to accept this manuscript for publication. Please make sure to upload a higher-resolution map (i.e., Figure 2). Additionally, I would like to encourage you to also release any code that can be used to reproduce this work (e.g., the figures you provide).

Thank you. We have adjusted the format of the references as requested in a prior email, and uploaded this version of the manuscript. We have also uploaded eps versions of the 8 figures, in a zip file under the 'Supplement' option.